# Teaching Sustainability in Planning and Design Education: A Systematic Review of Pedagogical Approaches

Hye Yeon Park *, Carlos V. Licon and Ole Russell Sleipness

Department of Landscape Architecture and Environmental Planning, Utah State University, Logan, UT 84322-4005, USA; carlos.licon@usu.edu (C.V.L.); ole.sleipness@usu.edu (O.R.S.)
* Correspondence: hyeyeon.park@usu.edu

**Abstract:** Sustainable development principles are being increasingly incorporated into university planning and design education. This paper evaluates how university planning and design programs teach sustainability and how these various approaches may influence future planners and designers. This systematic review quantitatively analyzes 5639 empirical research documents published from 2011 to 2020, including peer-reviewed papers and reports related to planning and design disciplines in higher education institutions. Key findings include differences in how planning and design curricula include and emphasize sustainability topics, as well as how various modes and teaching approaches correlate with sustainability values. This research offers a comprehensive understanding of how sustainable development approaches and teaching methods may influence how students and emerging professionals approach complex planning and design problems.

**Keywords:** sustainability; education for sustainable development (ESD); curriculum design; higher education; pedagogy; planning and design education

## 1. Introduction

Over the last decade, sustainability has emerged as a central theme of higher education institutions [1,2], along with the belief that education could be a vital aspect of strategy for sustainable development [3]. Accordingly, the United Nations decade of education for sustainable development emphasized incorporating the theory and practices of sustainable development into education [4]. These movements caused the advance of a new paradigm in the education field: education for sustainable development (ESD) [4–7]. Currently, ESD has become a contemporary consideration at all levels of education, including higher education [8]. ESD benefits school improvement and individual students, allowing them to ask critical questions about the status quo, clarify their values, and think systemically [9]. It also provides a meaningful real-world focus, helping students to be aware of the value of their lives and making schools improve themselves [10] (Barratt Hacking et al., 2010). Above all, students can gain direct sustainability experiences through ESD learning approaches [4]. These ESD features have helped researchers to recognize it as a vital way to attain sustainability [11,12].

In this paper, we have referenced the concept of ESD from the framework of UNESCO, which emphasizes encouraging learners' transformative action and structural changes by providing people with the skills to guarantee their living [13]. Additionally, we have accepted the argument in the Report of the World Commission on Environment and Development that "sustainability requires the enforcement of wider responsibilities for the impacts of decisions" [14]. Based on this foundation, we follow the purpose of ESD in nurturing future generations who can make informed decisions and take responsible action to resolve complex problems [15].

Universities act as significant educational conduits for the resolution of sustainable issues. Their primary roles, including research, teaching, and outreach, support sustainable

development and the goals of ESD at the institutional and community levels [16]. Over 600 universities worldwide have developed diverse educational programs focused on sustainability and sustainable development [17]. Notably, a wide array of fields has tried to incorporate ESD into academic areas, such as economics, environment, engineering, and the arts. Along with these trends, planning-related disciplines have increasingly embraced the concept of sustainability, which is also emerging as a new planning stream [18].

In the fields of planning and design, including urban planning, regional planning, landscape architecture, and urban design, sustainability is vital to address the development dilemmas of environmental protection, urban development, economic activity, and social expectations [19]. Design and planning decisions must consider a wide range of activities representing the goals of preservation, development, economic opportunities, social justice, and many others [19,20].

Given the increasing importance of design and planning-related professions and the long-term environmental effects of their decisions and tasks [21], the concept of sustainability into the teaching of planning and design programs should be integrated [22–24]. There is a significant body of research expanding the understanding of sustainability in planning and design education and identifying and selecting strategies to teach future planners and designers [25,26].

In order to identify ESD pedagogical approaches in planning and design courses, we conducted a two-step research procedure. First, preliminary background research on publications was performed to understand ESD approaches, experiences, and challenges comprehensively. Second, this paper examined how educators introduced the concept of sustainability in planning and design teaching and the teaching methods employed [27]. Specifically, we have examined teaching methods, pedagogical approaches, and the benefits and challenges of teaching sustainability.

*Background: Preliminary Research*

From October 2019 to December 2019, a preliminary publications review examined and synthesized forty publications from the Association for the Advancement of Sustainability in Higher Education (AASHE) using systematic review guidelines [28,29]. The preliminary study results revealed specific elements of educational experiences; for instance, sustainability appears as a specific dedicated course, or a theme in class. We assigned five groupings describing educational elements or dimensions when teaching sustainability: venue, subject, delivery, audience, and outcomes (Figure 1).

(1) **Venue** refers to where learning is happening. Venue can include the whole institution committed to teaching sustainability through programs, vision statements, or practice. More focused teaching approaches include programs, courses, projects, workshops, or field trips. (2) **Subject** describes the specific learning topic, theme, or course. (3) **Delivery** is the messenger in charge of teaching sustainability, which could include not only course instructors but also students, peer tutors, external community members, etc. (4) **Audience** describes the targeted learners. It can include students, the community at large, or a specific sector or industry, but in most cases, students are the target audience. (5) **Outcomes** identify and gather evidence of learning. Learning outcomes include tangible products, such as a project, and intangible outcomes, such as design thinking, collaboration skills, or even environmental awareness.

This literature review also identified 22 educational approaches. These approaches describe the pedagogical strategies implemented to enable learning. Among these strategies are collaborative learning, experiential learning, interdisciplinary studies, etc. The three dominant strategies implemented are interdisciplinary approaches (n = 7), transdisciplinary studies (n = 5), and competency-based approaches (n = 4). Table 1 combines these strategies with the educational elements or dimensions within which learning took place.

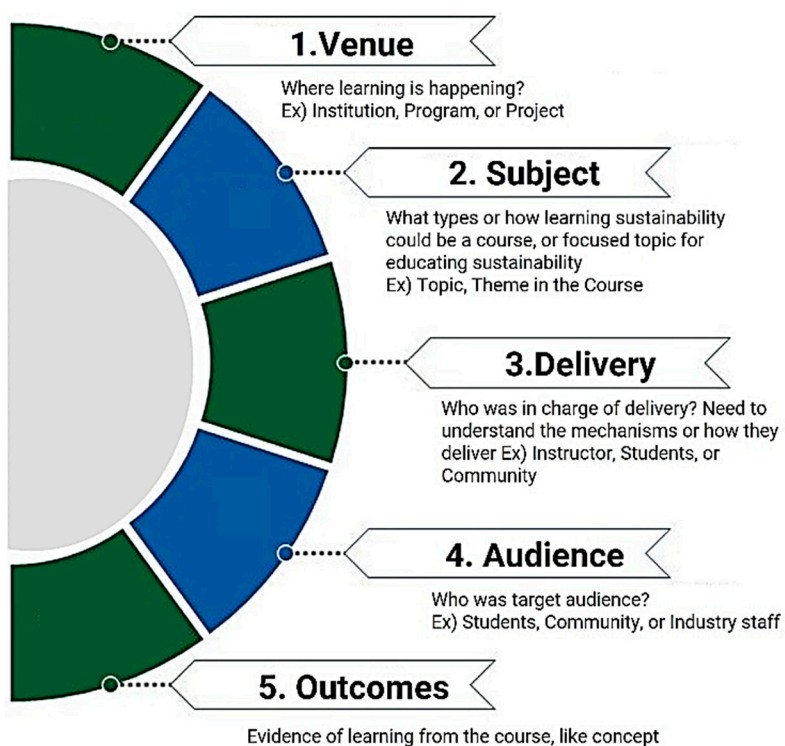

**Figure 1.** Five dimensions of sustainability educational experiences.

**Table 1.** Teaching approaches and dimensions in sustainability education.

| Rank | ESD Approaches in HE | Venue | Subject | Delivery | Audience | Outcome | Type of Disciplines |
|---|---|---|---|---|---|---|---|
| 1 | Interdisciplinary (n = 7) | 2 | 7 | 6 | 6 | 4 | Sustainability science [30], construction management/art and design [31], social science [32], general [33,34], sustainable development and management [35], combination of humanities, natural sciences, and social sciences [36] |
| 2 | Transdisciplinary (n = 5) | 2 | 4 | 2 | 2 | 2 | Sustainability science [12], education, ecology, environmental science, chemistry, economics and business, political science, psychology, etc. [26], sustainability science [30], combination of humanities, natural sciences, and social sciences [36], Social sciences and humanities perspectives [37] |
| 3 | Competencies (n = 4) | 2 | 4 | 2 | 3 | 3 | General [33,34], sustainable development and management [35], accounting and administration [38] |
| 4 | Transformative learning (n = 3) | x | 3 | 3 | 2 | 2 | General [39], general centering on a specific postgraduate program, economics, renewable energy, the development of affordable housing workspace, and local food production and processing [40], management [41] |
| 5 | Experiential learning (n = 2) | 2 | 2 | 2 | 2 | 2 | General [42], sustainable environmental Management [43] |
| 6 | Service learning (n = 2) | 2 | 2 | 2 | 2 | 2 | education, ecology, environmental science, chemistry, economics and business, political science, psychology, etc. [26], sustainable environmental management [43] |

**Table 1.** *Cont.*

| Rank | ESD Approaches in HE | Venue | Subject | Delivery | Audience | Outcome | Type of Disciplines |
|------|---------------------|-------|---------|----------|----------|---------|---------------------|
| 7 | Self-regulated learning (n = 2) | 1 | 2 | 2 | 2 | 1 | Construction management/art and design [31], combination of humanities, natural sciences, and social sciences [36] |
| 8 | Project-based learning (n = 2) | 1 | 2 | 1 | 2 | 2 | Construction management/art and design [31], urban planning [44] |
| 9 | Critical thinking/reflection (n = 2) | x | 2 | 2 | 2 | 2 | Social science [32], management [41] |
| 10 | Collaborative (n = 2) | x | 2 | 2 | x | 1 | General [31], sustainability science [30] |
| 11 | Problem and project-based learning (n = 1) | 1 | 1 | 1 | 1 | 1 | Sustainable environmental management [43] |
| 12 | Cross cultural (n = 1) | 1 | 1 | 1 | 1 | 1 | Construction management/art and design [31] |
| 13 | Learning landscape (n = 1) | 1 | 1 | 1 | 1 | 1 | General [42] |
| 14 | Human-centered design (n = 1) | 1 | 1 | 1 | 1 | 1 | Construction management/art and design [31] |
| 15 | Generalism, holism, and holarchism (n = 1) | 1 | 1 | 1 | 1 | x | Environmental science and policy [45] |
| 16 | Comprehensive (n = 1) | 1 | 1 | 1 | 1 | x | Environmental science and policy [45] |
| 17 | Interculturality (n = 1) | x | 1 | 1 | 1 | 1 | Sustainable development and management [35] |
| 18 | Reflective thinking (n = 1) | x | 1 | 1 | 1 | 1 | Social science [32] |
| 19 | Problem-based learning (n = 1) | x | 1 | x | 1 | 1 | Urban planning [44] |
| 20 | Case based learning (n = 1) | x | 1 | x | 1 | 1 | Urban planning [44] |
| 21 | Holistic approach (n = 1) | x | 1 | 1 | 1 | x | General centering on a specific postgraduate program, economics, renewable energy, the development of affordable housing workspace, and local food production and processing [40] |
| 22 | Transversality strategy (integrative approach) (n = 1) | 1 | 1 | x | x | 1 | Accounting and administration [38] |
| | **Sum of frequency of different dimensions** | 19 | 42 | 33 | 34 | 30 | |

This study found that in many cases, teaching sustainability implemented more than one approach. For instance, the transformative learning approach, which emphasizes students' critical skills, such as asking questions, finding reliable information, and critical thinking, was used along with a collaborative approach. This mix of strategies helps students address real-world issues from a holistic perspective [39,46]. Another notable result is that ESD approaches are frequently used in the "subject" dimension (n = 42). The "venue" dimension (n = 19) was the least preferred environment where sustainability teaching took place.

This preliminary review also documented the main outcomes resulting from these teaching efforts, including the development of comprehensive thinking [45], critical understanding of real-world issues [39], and in-depth learning [32,42]. In addition, the review

documented 23 challenges faced when teaching sustainability. The main challenges are the needed time and effort and the lack of sustainability awareness as the two most frequent and critical problems with existing educational barriers. Some cases faced community collaborating challenges, miscommunication among different fields, misunderstandings of sustainability, financial burdens, insufficient funding, etc. [39,46,47].

Based on the findings of the preliminary review, this study aims to narrow the scope and answer the following questions with respect to the disciplines of planning and design:

(1)    What educational approaches are being implemented in planning and design education?
(2)    What teaching methods are applied in these programs?
(3)    What are the benefits and challenges faced in teaching sustainability in planning and design?

## 2. Methodology

The study applied the preferred reporting items for systematic reviews and meta-analysis for protocols 2015 (PRISMA-P 2015). This research method facilitates the development and reporting of systematic review protocols to reduce arbitrary decision-making [48]. This study also compared ESD learning approaches in design and planning education with pedagogical approaches from the preliminary study described in the preceding sections. The objectives of this study are:

- to explore and characterize current ESD approaches in design and planning;
- to identify teaching modes in use and in combination with ESD pedagogical approaches;
- to identify issues and experiences in teaching sustainability; and
- to compare identified challenges with the results of the preliminary research.

### 2.1. Search Strategy

Data was collected from three different publication clearinghouses: AASHE, which offers quality resources related to sustainability curricula; the Education Resources Information Center (ERIC), which supports education research and information; and SCOPUS, a comprehensive high-quality scholarly database. The study limited the scope of the publications to those written in English and that were published from 2011 to 2020. The data screening and selection procedure followed the PRISMA-P 2015 guidelines suggested in Shamseer et al. [49] and McInnes et al. [50]. The review procedure includes identification, screening, eligibility, and inclusion stages. We followed the inclusion/exclusion method used by McInnes et al. [50]. Table 2 shows the inclusion and exclusion criteria used in this selection process.

**Table 2.** Inclusion and exclusion criteria of PRISMA-P 2015.

|   | Inclusion Criteria | Exclusion Criteria |
|---|---|---|
| 1 | Empirical research (survey or case study) | Nonempirical research (policy, theory, or methods) |
| 2 | Conducted in higher education | Not conducted in higher education |
| 3 | Managing course or program contents | Not discussed regarding the course or program contents |
| 4 | Applicable to design or planning-related education | Not applicable to design or planning education |
| 5 | Written in English | Written in other language types of English |

### 2.2. Data Collection and Analysis

After using the inclusion and exclusion criteria, this study applied different combinations of terms for collecting samples as listed in Table 3. Since data from AASHE includes sustainability by organizational definition we used a different set of keywords focusing on "environment" and "education." The resulting selection included 753 citation

records in publications on research and higher education curricula. The data from ERIC covers education, and thus the keywords "environmental" and "sustainability" were used, with specific criteria, such as" peer-reviewed", "journal article", and "higher education". The results presented 660 items. For the SCOPUS search, we applied keywords, such as "environmental" and "sustainability" and "education" with "curriculum" or "course" and "university" or "college" or "higher education".

**Table 3.** The data collection process.

| Database | Applied Inclusion Criteria | Result |
|---|---|---|
| **AASHE** **(n = 753)** | • Content-type: case studies and publications, sustainability topic: curriculum and research<br>• Year posted: 2011–2020<br>• Applying related to disciplines (architecture, behavior sciences, design, education, environmental studies, social sciences, arts, sustainability studies, urban, community, and regional planning) | • Environment + Education: 103<br>• Environment + Educational: 104<br>• Environmental + Education: 164<br>• Environmental + Educational: 163<br>• Environment + Environmental + Education + Educational: 80<br>• Environmental AND Sustainability AND Education: 139 |
| **ERIC** **(n = 660)** | • Full text available on ERIC<br>• Year posted: 2011–2020<br>• Applying: higher education level | • Environment AND Sustainability AND (Design OR Planning): 49<br>• Environment AND Sustainable AND (Design OR Planning): 51<br>• Environmental AND Sustainability AND (Design OR Planning): 35<br>• Environmental AND Sustainable AND (Design OR Planning): 31<br>• (Environment OR Environmental) AND (Sustainability OR Sustainable) AND (Design OR Planning): 110 |
| **SCOPUS** **(n = 4226)** | • Search within article title, abstract, and keyword, open access<br>• Year posted: 2011–2020<br>• Applying related to disciplines (social sciences, environmental science, earth and planetary sciences, agricultural and biological sciences, and arts and humanities) and final publication stage | • Environment AND Sustainability AND Education AND Design OR Planning: 250<br>• Environment AND Sustainability AND Educational AND Design OR Planning: 86<br>• Environment AND Sustainable AND Education AND Design OR Planning: 408<br>• Environment AND Sustainable AND Educational AND Design OR Planning: 154<br>• Environmental AND Sustainability AND Education AND Design OR Planning: 334<br>• Environmental AND Sustainability AND Educational AND Design OR Planning: 108<br>• Environmental AND Sustainable AND Education AND Design OR Planning: 489<br>• Environmental AND Sustainable AND Educational AND Design OR Planning: 180<br>• Environment OR Environmental AND Sustainability OR Sustainable AND Education OR Educational AND Design OR Planning: 973<br>• Environmental AND Sustainability AND Education: 1245 |

As shown in Figure 2, the identification stage produced a total of 5639 hits. After the identification stage, we removed duplicates and applied exclusion criteria, removing articles written in languages other than English, or those not focused on higher education. This step reduced the set to 1951 items. Further filters eliminated items not related to the disciplines of planning and design. The selection emphasis was on course content, with few additional papers covering case studies or institutional changes [51–53]. At this

point, 94 publications survived the cut. Finally, we selected 41 papers for the systematic review that were closely related to planning and design-related education. The focus on searching for empirical or case study-based research regarding sustainability teaching methods and approaches resulted in the elimination of some nonempirical research and some specific reports with reliability issues. However, the selected samples were sufficient for the analysis in the study.

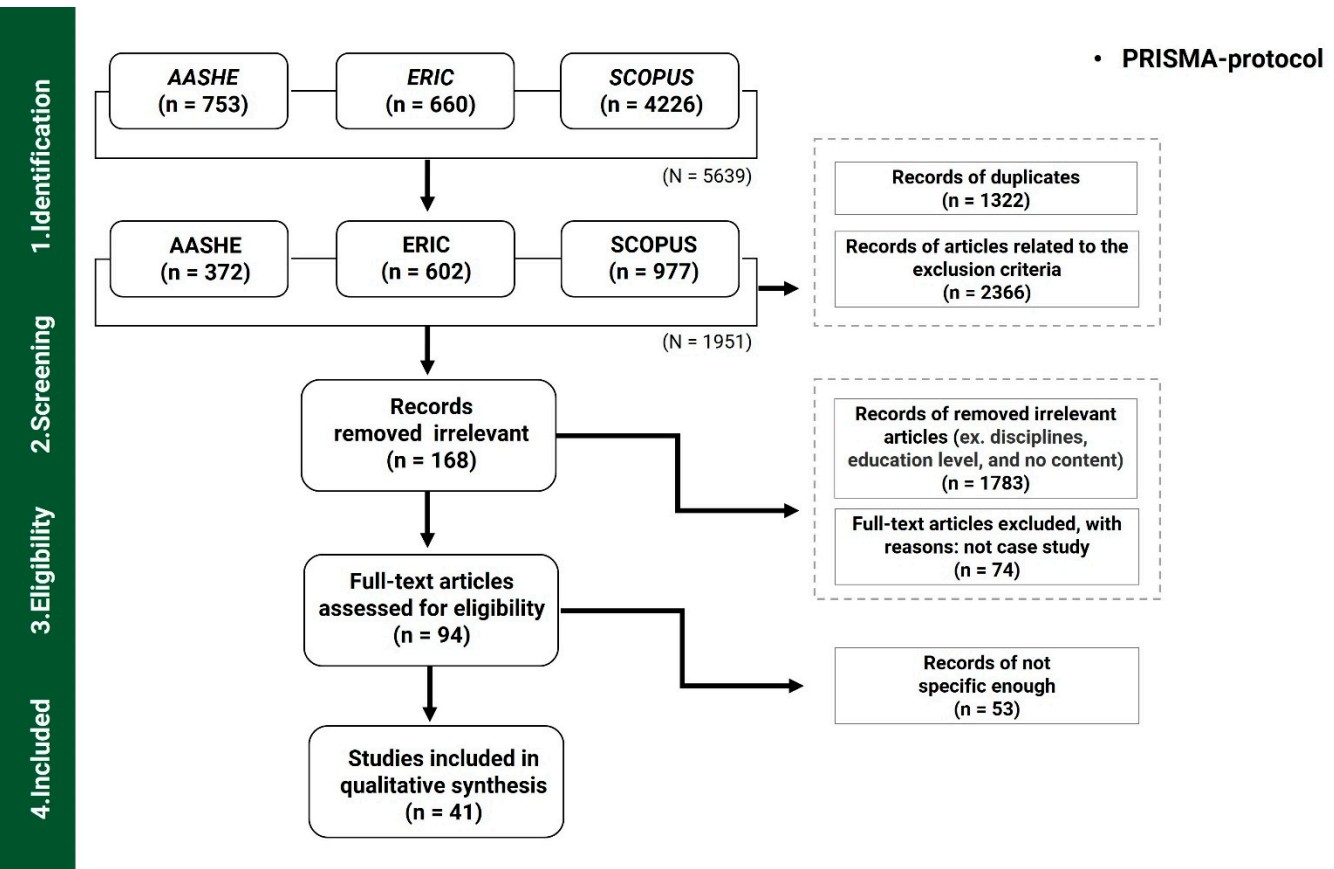

**Figure 2.** The procedure of PRISMA-protocol.

### 3. Results

After analyzing 5639 publications, this study selected a total sample of 41 articles looking at higher education institutions with planning or design programs in 37 countries, with the greatest number of institutions found in the United States (39.0%), followed by Southern Europe (14.6%) and Asia (14.6%). Western Europe, the UK, and Australia each comprised less than 10 percent of the sample analyzed. The selected articles were published in 20 journals covering sustainability (SUS), education (EDU), environment (ENV), ecology (ECO), and design (DES). Table 4 shows that the education category covers most journals (n = 19).

**Table 4.** Category of reviewed journals.

| Category | Journals |
|---|---|
| Sustainability | Sustainability (6)<br>Current Opinion in Environmental Sustainability (1)<br>Michigan Journal of Sustainability (1) |
| Environment | Journal of Cleaner Production (4)<br>Journal of Future Studies (2)<br>Journal of Green Building (1) |
| Environmental Sciences | Journal of Integrative Environmental Sciences (1)<br>Journal of Environmental Studies and Sciences (1) |
| Ecology | Habitat International (1)<br>GAIA-Ecological Perspectives for Science and Society (1) |
| Higher Education | International Journal of Sustainability in Higher Education (5)<br>Higher Education Pedagogies (2)<br>Journal of Problem-Based Learning in Higher Education (2) |
| Education/Pedagogies | International Journal of Teaching and Learning in Higher Education (1)<br>Scandinavian Journal of Educational Research (1) |
| Environmental education | Environmental Education Research (4) |
| Natural resource education | Journal of Natural Resources and Life Sciences Education (1)<br>Solar Energy (1) |
| Biology education | Journal of Biological Education (2) |
| Design | The Design Journal (1) |

*3.1. ESD Approaches in Planning and Design Education*

To identify ESD approaches when teaching sustainability in design and planning education, we examined the teaching methods used, as well as the benefits and challenges described in the revised articles. Compared to the preliminary research results, the findings after the PRISMA protocol showed 24 ESD learning approaches with specific purposes for design and planning education (Table 5). In many cases, the pedagogical strategies implemented consist of a combination of approaches. For instance, some cases combine action-oriented with transformative approaches [54], while in other examples, courses combine problem-based and project-based learning approaches to provide students with practical experiences in community service projects [43,55]. The following descriptions offer some examples of eight innovative ESD learning approaches that are not found in the preliminary research stage.

**Table 5.** 24 ESD approaches in planning and design education courses.

| 24 ESD Approaches in Planning and Design Courses | | |
|---|---|---|
| 1 | Action-oriented (n = 7) | Action competence and transformative learning, action research, action-oriented transformative pedagogical approach, active learning constructivist approach, active learning (n = 3) |
| 2 | Interdisciplinary (n = 6) | Interdisciplinary and crosscultural setting, interdisciplinary approach, interdisciplinary education, interdisciplinary (urban planning education in post-social transitional countries (UPEPSTCs)), interdisciplinary (multidisciplinary) (n = 2) |
| 3 | Problem-based learning (n = 6) | Problem- and project-based learning (PPBL), problem-based learning (n = 3), problem solving approach (n = 2) |
| 4 | Project-based learning (n = 5) | Project-based learning (PBL) and service learning (SL), project-based learning (PBL) (n = 4) |

**Table 5.** *Cont.*

| | 24 ESD Approaches in Planning and Design Courses | |
|---|---|---|
| 5 | Experiential learning (n = 4) | Experience-based learning, experiential learning (n = 3) |
| 6 | Place-based learning (n = 4) | Place-based learning, Place-based education (PBE) and experiential learning, Place-based education (n = 2) |
| 7 | Participatory action research (n = 3) | Participatory (sustainable architectural design studios (SADS), participatory action research (n = 2) |
| 8 | Service learning (n = 3) | service learning approach (n = 3) |
| 9 | Transformative (n = 3) | Transformative learning (n = 3) |
| 10 | Crosscultural (n = 2) | Multicultural education, crosscultural collaboration |
| 11 | Collaborative (n = 2) | Collaborative learning, Collaborative action research |
| 12 | Integrative (n = 2) | Integrative approach (n = 2) |
| 13 | Case-based learning | Case method teaching |
| 14 | Competency-based | Competency-based approach |
| 15 | Experimental studio | Experimental green design studio |
| 16 | Future-oriented | Future-oriented learning |
| 17 | Holistic approach | Holistic and human rights-oriented approach |
| 18 | Learning network | Learning network approach |
| 19 | Performance-oriented | Performance-oriented architecture |
| 20 | Self-regulated | Self-regulated learning |
| 21 | Solution-oriented | Solution-oriented sustainability learning (SOSL) |
| 22 | The burn model | Burn model sustainability pedagogy |
| 23 | Three-fold framework | A 'three-fold' framework of activities on the environment (self-reported outcome) |
| 24 | Transdisciplinary | Transdisciplinary approach |

### 3.1.1. Experimental Studio (Green Design Studio)

The experimental studio includes teaching sustainable green methods of design and construction through design projects and living lab experiments. This learning approach requires students to design given extreme wind conditions and conduct a workshop in a living lab situation to experiment with and test environmental solutions, such as energy efficiency [56].

### 3.1.2. The Burn Model of Sustainability Pedagogy

The burn model "integrates ecological design, systemic and interdisciplinary learning, multiple perspectives, an active and engaged learning process, and attention to place-based learning" [57]. It focuses on applying sustainability pedagogy from diverse perspectives with practical suggestions for teaching sustainability. Teaching modes include large or small group discussions, meeting guest speakers, field trips, and journal writing [57].

### 3.1.3. A Three-Fold Framework of Activities on the Environment

This approach aims to "promote multiple learning outcomes to enable students (of any age) to participate in various learning experiences." The three-fold framework focuses on education for the environment and in/from the environment, including basic knowledge, investigation, environmental concerns, values, and attitudes. Specifically, it involves lectures, fieldwork, investigations, data analysis, class presentations, discussions on human impact on the environment, ethical issues and questions, etc. [58].

### 3.1.4. Learning Networks

Learning networks pursue "bottom-up approaches as well as self-organization, while the organizational, educational, and technological components are activated to encourage self-directed learning processes jointly" [59].

It emphasizes open communication and supports "the transition of the educational system that would be difficult to accomplish within traditional organizational frameworks" [59]. Teaching includes essays, discussion forums, writing research proposals and group presentations, collaboration with regional players, and a virtual seminar [59].

### 3.1.5. Future-Oriented Learning

Future-oriented learning pursues "the experimental-innovative game-based futures curriculum design" and aims to "participate, facilitate, collaborate, and play with students in the classroom world, like less lecture, more play" [60]. This approach emphasizes a game's strength in spatial planning and understanding sustainability. Players or learners can "interact with artifacts, test ideas, attempt their strategies, and adapt to changing conditions as the game progresses to fulfill their goals" [60]. The teaching method consists mainly of the contents of the games, such as exploring images of the future, collaborative activities, mapping the future, graphical visualization of direct and indirect results according to future development, making headline news, and having debates [60].

### 3.1.6. Performance-Oriented Learning

The performance-oriented approach emphasizes "an interdisciplinary approach to establishing adequate starting positions for tackling compound sustainability problems through design" [61]. It connects design thinking and systems thinking to address a broad scope of actors and stakeholders and also pursues expanding the remit far beyond human-centric design. Teaching content includes interviews with locals and visitors, collaboration with stakeholders and students, field trips, and analysis [61].

### 3.1.7. Solution-Oriented Learning

The solution-oriented learning approach consists of "competencies-based and experiential learning, which allows students to learn while transforming" [62]. This approach aims to change passive learning to active, transformative, participatory, and project-based learning. It offers students the opportunity to learn about informed sustainability problems and build the capability to solve them. During the course, instructors offer students an overview of sustainability problems, involving collaborations with experts and stakeholders, field trips, making products, such as plans, policies, reports, and webpages, developing scenarios and visualizations of urban futures, boot camps, small group exercises, and incorporating external facilitators [62].

### 3.1.8. Participatory Action Research (PAR)

PAR approach is a design studio with participatory and social features. It pursues more practical knowledge to complement theoretical knowledge by integrating real sustainability issues into design projects. The course content in PAR involves small group projects and discussions, field trips, presentations, workshops with experts, critical design approaches, concept mapping, and reflective journals [63–65].

### *3.2. ESD Approaches with Methods*

One of the study's objectives is to understand how different teaching methods or modes support different ESD pedagogical approaches. This combination of tactics (teaching modes) with strategies (approaches) offers a valuable framework to articulate and integrate different ways to teach sustainability in design and planning education.

Figure 3 shows the frequency of use of each teaching mode for each of the 24 approaches identified. The top five pedagogical approaches recognized in design and planning education are teaching through action-oriented approaches (n = 7), interdis-

ciplinary approaches (n = 6), problem-based learning (n = 6), project-based learning (n = 5), and experiential learning (n = 4). Overall, pedagogical approaches seem to be more focused in teaching sustainability through practice and learning-by-doing activities [63–65]. These practice and experiential strategies rely on group projects, collaboration with local communities, NGOs, industries, and other institutions, and sharing through group presentations [16,55,59,66–70]. Teaching through lectures is still among the dominant teaching modes.

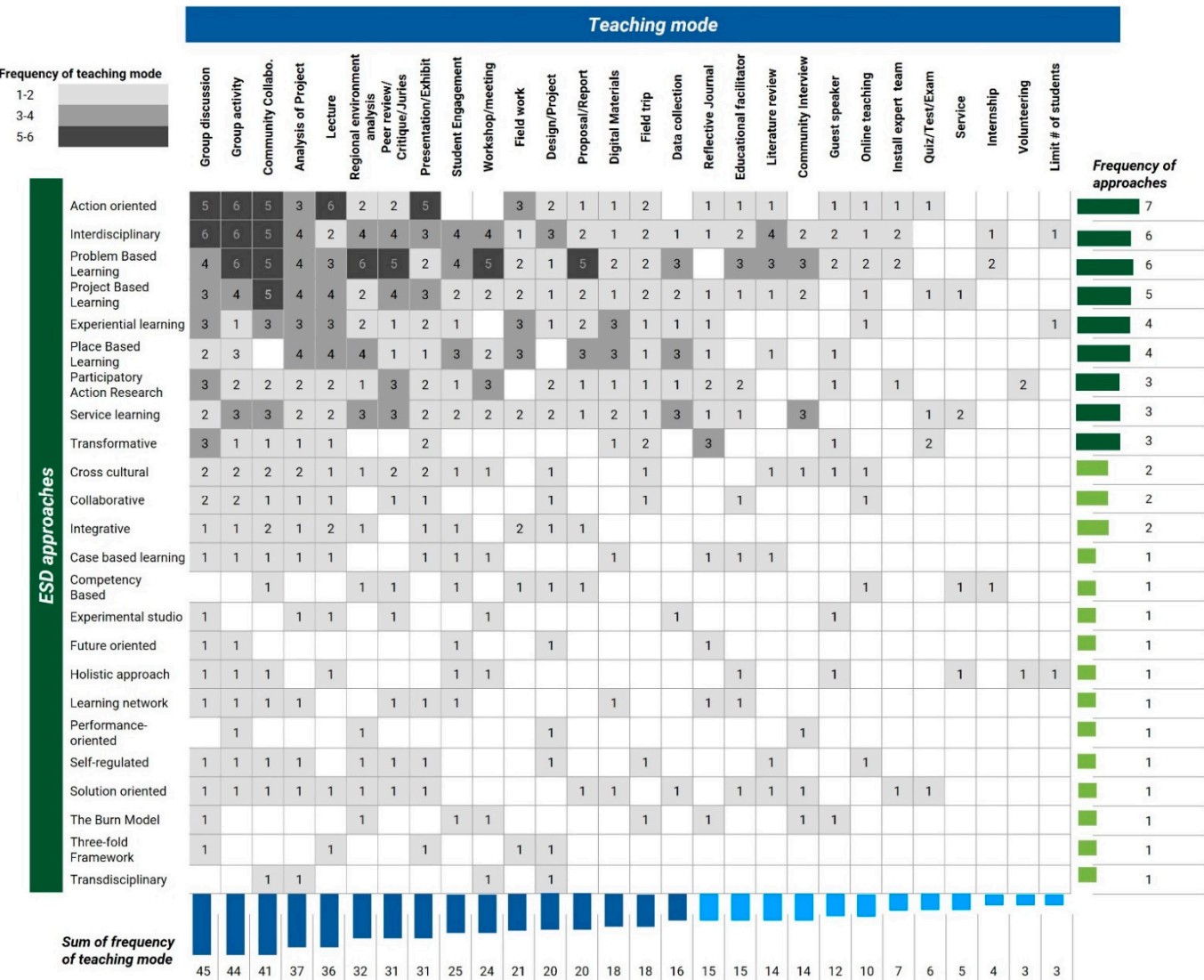

**Figure 3.** The framework of ESD approaches and teaching modes or methods in planning and design courses.

### 3.3. Benefits and Challenges of Teaching Sustainability

This study identified 22 benefits, strengths, and positive outcomes described by the authors of these articles after their experience in teaching sustainability (Figure 4). These authors documented benefits through surveys, workshops, or direct feedback from students. The most notable benefits are developing problem-solving skills, obtaining critical thinking, development of design and planning abilities, and building collaboration skills [32,71–74], shown the blue color in Figure 4. Additionally, students addressed complex real-life issues during these courses. The study assumes that considering real and complex issues through sustainability courses can help future planners and designers to develop stronger critical thinking skills.

- **22 Strengths or benefits after applying teaching approaches in the study**

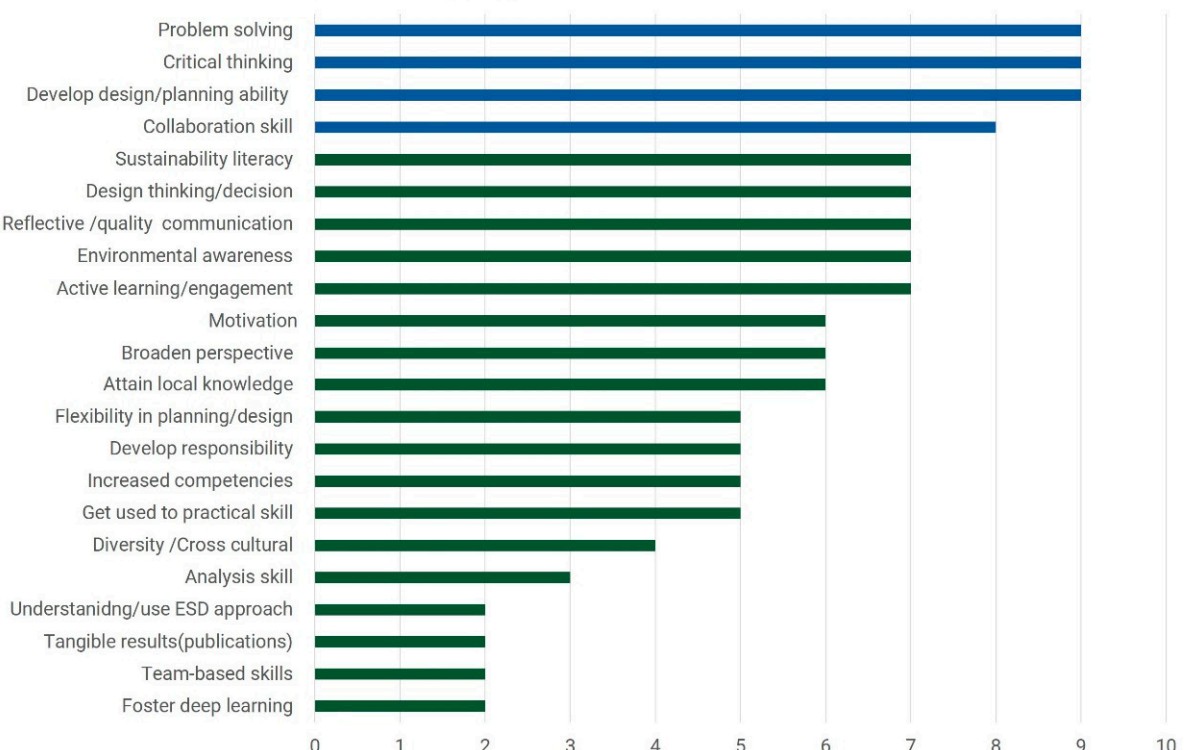

**Figure 4.** Benefits of teaching sustainability in planning and design education.

Figure 4 shows that teaching sustainability also helps students to develop design and planning abilities, such as design thinking [75,76]. The implementation of sustainability teaching in planning and design also faces some challenges and restrictions. Figure 5 shows the main issues, barriers, and challenges that instructors and students have faced while employing ESD approaches in their courses. The need for significant amount of time and effort to develop learning opportunities to teach sustainability is the most significant problem and need specific guidance stated in the articles reviewed, shown the blue color in Figure 5. Higher demands are placed on instructors in the classroom, such as requiring a lot of time and effort, requiring specific guidelines, and using environmental restrictions. Team- or project-oriented difficulties frequently appeared while conducting ESD in design and planning education. Ultimately, the complex and long-term issues that define the core of sustainability views also affect how sustainability can be taught. Efforts to incorporate long-term views, interdisciplinary perspectives, or participatory processes, require longer term studies, time to discuss and assimilate issues, and sometimes a more supportive administrative structure to carry these efforts to successful outcomes.

- **21 challenges or weaknesses after applying teaching approaches in the study**

**Figure 5.** Challenges of teaching sustainability in planning and design education.

## 4. Discussion

As found in our exploratory review of publications, many studies that expand the understanding of sustainability do not explain the relationship between teaching methods and ESD approaches. Our current study connects teaching methods and pedagogical approaches in the classroom by recognizing two relevant topics for discussion. First, the literature review confirms that instructors' responsibility for the course is vital to teaching sustainability. This result might suggest that teachers need more training, experience, or knowledge of the complex learning process in teaching sustainability. Second, as with the preliminary research findings (step 1), step 2 also indicates that the complexity of teaching sustainability calls for utilizing more than one teaching method and ESD approach, and that these approaches should be innovative, evolved, and specific. Therefore, implementing complex ESD approaches should involve several considerations, including well-designed learning environments, resources, and careful support from institutions, and educators' sufficient capability might help to teach sustainability better in planning and design education. An important finding is a clear link between ESD approaches and teaching methods. The current results revealed how sustainable development approaches and teaching methods contribute to students' ability to solve complex planning and design problems. Our results thus confirm the vital role that ESD approaches can play in improving the learning environment and the required capabilities of future planners and designers. Furthermore, it might suggest that existing traditional courses teaching sustainability should undergo major revision to achieve positive outcomes through ESD approaches.

## 5. Conclusions

The current study aimed to investigate and characterize contemporary ESD approaches in design and planning fields, to understand teaching modes, and to combine these approaches with teaching modes to build a framework. In addition, we identify a variety of issues and experiences in teaching sustainability.

The research findings clearly show that applying ESD approaches benefits design and planning students, even though it requires intentional effort and flexibility on the part of both faculty and students. Instructors play a critical role in successfully integrating ESD approaches into curricular and course content through their responsibility. Consequently, instructors need to understand the complex concepts of sustainability, be open to integrat-

ing new educational modalities, and master ESD approaches and teaching methods to offer specific guidance and solutions-based processes. Next, the study results encourage planning and design programs to be up to date on ESD approaches and related teaching methods. For instance, some existing teaching methods in planning and design education may be too simple for teaching complex sustainability concepts. We suggest that integrating teaching approaches and modes from different disciplines may provide current ideas on addressing complex social issues. Additionally, collaborative efforts from institutions, faculty, students, and the community may augment interdisciplinary approaches.

Our research on planning and design disciplines contribute to a comprehensive understanding of how the disciplines teach sustainability. However, this broad research scope and area made it challenging to organize the research results regarding ESD approaches and teaching modes. Further studies may establish specific guidance on integrating innovative teaching methods from other disciplines into planning and design.

**Author Contributions:** The research was designed and supervised by C.V.L.; the data were collected, analyzed, and written by H.Y.P.; finally, checked and revised by O.R.S. All authors have read and agreed to the published version of the manuscript.

**Funding:** This research received no external funding, and the APC (Article Processing Charge) was funded by OA Funding Initiative Award, Utah State University (Utah, USA).

**Institutional Review Board Statement:** Not applicable.

**Informed Consent Statement:** Not applicable.

**Conflicts of Interest:** The authors declare no conflict of interest.

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
