# Peer review of "Teaching Sustainability in Planning and Design Education: A Systematic Review of Pedagogical Approaches"

_sustainability, doi:10.3390/su14159485_

Round 1

Reviewer 1 Report

 My feeling is that the paper needs some revisions before it can be published.    I don't understand table 1. Venue, subject etc. seem odd criteria and aren't explained in the text.    Line 192 "other language types of English"    Line 197 and 198 The authors don't explain how they got from 94 publications to 41 papers for the systematic review.    In Figure 3 the greyscale is very hard to read, plus the "amount of ESD cases" section of the table is not easy to understand.    Section 3.3. is lacking in specific examples of the different strengths/benefits and challenges/weaknesses. This means that the authors' promise in the abstract that the review would help the reader understand how teaching methods may influence how students approach complex planning and design problems is not achieved.   

If I were a lecturer reading this review with the hope that I would learn how to create new planning and design education then I would be disappointed. If I were someone interested in being sign-posted to other papers to read, then I would be happy. 

Author Response

Dear Reviewer,

I appreciate your patience and am sorry for the late response. The suggestions offered by the reviewers have been beneficial, and we appreciate your valuable comments on revising the paper.

We have included the reviewer comments immediately after this note and responded to them individually, indicating exactly how we addressed each concern or problem and describing the changes we have made.

We hope the revised manuscript will better suit the Journal but are happy to consider further revisions, and we thank you for your continued interest in our research.

Sincerely,
Hye

Comments and Suggestions for Authors

 Point 1: My feeling is that the paper needs some revisions before it can be published. I don't understand table 1. Venue, subject etc. seem odd criteria and aren't explained in the text.   

Response 1: We appreciate the time spent carefully reviewing the manuscript and providing your insightful comments and valuable feedback on our paper. Using the dimensions in Table 1, like venue, subject, delivery, etc., we wanted to systematically understand how sustainability is happening in the literature review. In the manuscript, we added additional explanations. We believe this additional information helps readers understand it clearly. The changes in the text appear in red type in the revised paper.

 Point 2: Line 192 "other language types of English”, Line 197 and 198 The authors don't explain how they got from 94 publications to 41 papers for the systematic review.   

Response 2: Thank you so much for catching these, which we have now corrected. Also, we added a screening process for how we got from 94 publications to 41 papers.

 Point 3: In Figure 3 the greyscale is very hard to read, plus the "amount of ESD cases" section of the table is not easy to understand.   

Response 3: Both you and the other reviewer commented on this table, so we are grateful to know that our current approach requires some rethinking. We have considered both solutions and decided to improve Figure 3 by re-organizing frequency and adding Table 5 for readers’ understanding. We believe this sets the information out clearly and comparatively and is a format that readers will readily return to when seeking information on the manuscript’s scribes and production.

 Point 4: Section 3.3. is lacking in specific examples of the different strengths/benefits and challenges/weaknesses. This means that the authors' promise in the abstract that the review would help the reader understand how teaching methods may influence how students approach complex planning and design problems is not achieved.   

Response 4: Thank you for your assessment. We agree that adding specific examples of the different strengths/benefits and challenges/weaknesses would be more helpful for readers, and we have taken your advice. You can check related figures (Figure 4,5) in the previous manuscript, and we added specific examples in the manuscript for accurate delivery of meaning.

 Point 5: If I were a lecturer reading this review with the hope that I would learn how to create new planning and design education then I would be disappointed. If I were someone interested in being sign-posted to other papers to read, then I would be happy. 

Response 5: We understand the reviewer's concern regarding a lack of the practical part of ‘how to create new planning and design education.' However, we would like to point out that this study focuses on a comprehensive understanding of sustainable development approaches and teaching methods by exploring trends and experiences. Also, the current research is an initial step before presenting practical parts.

Author Response

Dear Reviewer,

I appreciate your patience and apologize for the late response. The suggestions offered by the reviewers have been beneficial, and we appreciate your valuable comments on revising the paper.

We have included the reviewer comments immediately after this note and responded to them individually, indicating exactly how we addressed each concern or problem and describing the changes we have made.

We hope the revised manuscript will better suit the Journal but are happy to consider further revisions, and we thank you for your continued interest in our research.

Sincerely,
Hye

--------------------------------

Point 1: In the Abstract, you mention that “This research offers a comprehensive understanding of how sustainable development approaches and teaching methods may influence how students—and emerging professionals—approach complex planning and design problems”. This aim is not reflected in the Introduction (lines 70-81) where you describe the research methodology. Also, no mention about this in your Conclusion. You may consider removing or strengthen the supporting arguments/points on student’s skills and abilities.

Response 1: We appreciate the time spent carefully reviewing the manuscript and providing your insightful comments and valuable feedback on our paper. Based on your comments, we made the changes and improved the discussion and conclusion parts.

Point 2: A suggestion is to briefly explain what “Planning and design education” is, especially for readers who are not aware of this field. This could probably be included in the Introduction.

Response 2: Thank you for reminding us how important it is to explain the planning and design to readers. We agree with your comment, and we improved it.

Point 3: A suggestion is that the paper would have been more interesting if you had included the correlation of: type of school, academic department and discipline in “The preliminary research result, 2019” – Table 1.

Response 3: Thank you! We found your comments extremely helpful. Both you and the other reviewer commented on this table, so we are grateful to know that our current approach requires some rethinking. We have considered both solutions and decided to improve the table by adding a type of discipline with references and the frequency of ESD approaches.

Point 4: The two steps of the research procedure should be more tightly coupled in the discussion of the results findings and the conclusion.

Response 4: Thank you for this excellent observation. We improved the discussion and the conclusion for a more structured and clear manuscript. The revisions are marked in red in the revised paper.

Point 5: In the paper keywords, consider changing “Curriculum” to “Curriculum Design”.

Response 5: Thank you! We agree with your comment and have revised it.

Point 6: The Conclusion could be enhanced to incorporate the main concepts investigated, aims of the research and summarized results.

Response 6: We thank the reviewer for pointing this out. We have revised this more clearly.

Point 7: Line number: correction/suggestion

12, 13: are the documents and papers on higher education only? Consider specifying that.

14: consider editing: “design disciplines’ …”

50: consider changing “economy” to “economics”

63-69: consider improving the coherence of concepts.

68: if this is the “first step”, are you proposing a number of subsequent steps?

71: please clarify “teaching in the education field” – it is a rather confusing statement

70-81: consider improving the English language of this paragraph so that the expression of the steps and the meanings are clearer. Try avoiding repetitive words

91: consider rephrasing “how learning sustainability could be a course”. What is “learning sustainability”?

92: consider clarifying “who was in charge of delivery”. Delivery of a course that already exists? The impression is that you are talking about courses to be designed. Are students responsible for course delivery? If you are talking about student peer tutoring, you need to be very specific on this.

95: consider clarifying “how sustainability is happening in the literature review”. What do you mean “happening”?

97: consider rephrasing Figure 1: “Five dimensions of sustainability educational experiences”

110-112: consider improving the English language of this paragraph

141: consider changing “Methods” to “Methodology”

176: In Table 2, check the 2nd criterion (inclusion – exclusion)

308-329: Conclusion needs significant improvement in the English language used – confusing conclusion with syntax errors

Response 7: We thank the reviewer for these careful observations. We have carefully reviewed the entire manuscript and adjusted every relevant sentence to avoid dangling modifiers and clarify our meaning. Revised sentences are marked in red in the revised manuscript. In the case of your comment, “68: if this is the “first step,” are you proposing a number of subsequent steps?” we would like to point out that the intention of using the word “first step” was to show our research will be an “initial step” in advance of future research, not propose subsequent steps in this manuscript. However, we revised the word and lengthened the explanation for accurate clarification.

Reviewer 3 Report

The paper discusses an issue that contributes to the knowledge in Sustainability in Planning and Design Education. However, I have raised a few comments I believe would strengthen the manuscript before publication:

 1. ABSTRACT. Some claims are not fully demonstrated in the manuscript. For example, ‘This research offers a comprehensive understanding of how sustainable development approaches and teaching methods may influence how students—and emerging professionals—approach complex planning and design problems’ (16-17). These approaches should be further explained in the results and conclusion sections. The authors focus on the quantitative data but not on the content and main contributions of those approaches.

 2. RESULTS. Why did the authors put together Asia and Southern Europe? In the statement  ‘Southern Europe and Asia (n=6, 14.6%) were the second-highest numbers’ (218).  This grouping does not seem to be coherent.

English language editing is required, please read below:

 What are teaching methods applied in planning…? (139)  What teaching methods are applied?

‘Selecting contents type was publications..’ (183)

‘because they are broad range or focus on institutions' changes ..’ (196-197)

‘Since we searched empirical research regarding ..’ (205)

‘they were too broad term and required…’ (212)

‘studies gained related ideas through surveys..’ (276)

‘Notably, the time and effort needed’ (291) the time

‘this broad research scope proved challenging for organize …’ (324)

‘for organizing how ESD approaches and enhance teaching methods are incorporated’ (326-327)

Table 2. in Inclusion and Exclusion criteria number 2 is the same in both cases ‘Conducted in higher education’. I guess ‘Exclusion’ should be ‘NOT conducted in higher education’

Figure 2. Please, check the text, several examples are confusing: ‘Records of exclusion articles’ (?)

Figure 5. Title ‘Challenges or weakness’ – or weaknesses?

Author Response

Dear Reviewer,

I appreciate your patience and am sorry for the late response. The suggestions offered by the reviewers have been beneficial, and we appreciate your valuable comments on revising the paper.

We have included the reviewer comments immediately after this note and responded to them individually, indicating exactly how we addressed each concern or problem and describing the changes we have made. 

We hope the revised manuscript will better suit the Journal but are happy to consider further revisions, and we thank you for your continued interest in our research.

Sincerely, 
Hye

Comments and Suggestions for Authors

The paper discusses an issue that contributes to the knowledge in Sustainability in Planning and Design Education. However, I have raised a few comments I believe would strengthen the manuscript before publication:

Response: We appreciate the time spent carefully reviewing the manuscript and providing your insightful comments and valuable feedback on our paper. We have revised it to reflect your suggestions and highlighted the changes within the manuscript. Here is a point-by-point response to your comments. In what follows, your comments are in black, and the authors ‘responses are in red.

Point 1: ABSTRACT. Some claims are not fully demonstrated in the manuscript. For example, ‘This research offers a comprehensive understanding of how sustainable development approaches and teaching methods may influence how students—and emerging professionals—approach complex planning and design problems’ (16-17). These approaches should be further explained in the results and conclusion sections. The authors focus on the quantitative data but not on the content and main contributions of those approaches.

Response 1: Thank you! We found your comments extremely helpful and have revised them accordingly. We agree that the results and conclusion section would be better if it included the main contributions we mentioned in the abstract. Thus, we made the changes and improved sections based on your comments.

Point 2: RESULTS. Why did the authors put together Asia and Southern Europe? In the statement ‘Southern Europe and Asia (n=6, 14.6%) were the second-highest numbers’ (218).  This grouping does not seem to be coherent.

Response 2: We apologize for the unclear division. We did not intend to group “Southern Europe and Asia,” and it intended that Southern Europe (n=6, 14.6%) and Asia (n=6, 14.6%).  We have modified it and added the figure and hope it is now clear.

Point 3: English language editing is required, please read below:

What are teaching methods applied in planning…? (139) What teaching methods are applied?

‘Selecting contents type was publications..’ (183)

‘because they are broad range or focus on institutions' changes ..’ (196-197)

‘Since we searched empirical research regarding ..’ (205)

‘they were too broad term and required…’ (212)

‘studies gained related ideas through surveys..’ (276)

‘Notably, the time and effort needed’ (291) the time

‘this broad research scope proved challenging for organize …’ (324)

‘for organizing how ESD approaches and enhance teaching methods are incorporated’ (326-327)

Response 3: Thank you so much for catching these glaring and confusing errors, which we have now corrected. We have carefully reviewed the entire manuscript and adjusted every relevant sentence to avoid dangling modifiers and clarify our meaning. Revised sentences are marked in red in the revised manuscript.

Point 4: Table 2. in Inclusion and Exclusion criteria number 2 is the same in both cases ‘Conducted in higher education’. I guess ‘Exclusion’ should be ‘NOT conducted in higher education’

Response 4: We thank the reviewer for pointing this out. We have revised this.

Point 5: Figure 2. Please, check the text, several examples are confusing: ‘Records of exclusion articles’ (?)

Response 5: We thank the reviewer for pointing this out. We have revised this more clearly, like “Records of articles related to the exclusion criteria.”

Point 6: Figure 5. Title ‘Challenges or weakness’ – or weaknesses?

Response 6: Thank you for this excellent observation.  We’ve corrected the typo.

Round 2

Reviewer 2 Report

The authors have addressed all points mentioned in the first round of the review and have made all necessary changes and improvements.